# Essential Tremor Severity Assessment Using Handwriting Analysis and Machine Learning

**DOI:** 10.3390/s26010244

**Published:** 2025-12-31

**Authors:** Jose Ignacio Sánchez Méndez, Elsa Fernandez, Alberto Bergareche, Karmele Lopez-de-Ipina

**Affiliations:** 1NTT DATA EU & LATAM USA Branch Inc., 4100 North Fairfax Drive, Suite 810, Arlington, TX 22203, USA; 2EleKin Research Group, University of the Basque Country (EHU), 20018 Donostia, Spain; 3Department of Computational Science and Artificial Intelligence, University of the Basque Country (EHU), 20018 Donostia, Spain; elsa.fernandez@ehu.eus; 4Movement Disorders Unit, Department of Neurology, University Hospital Donostia, Paseo Doctor Begiristain, 109, 20014 Donostia, Spain; abergare@proton.me; 5Department of Systems Engineering and Automation, University of the Basque Country (EHU), 20018 Donostia, Spain; 6Department of Psychiatry, University of Cambridge, Herchel Smith Building Forvie Site Robinson Way Cambridge, Cambridge CB2 0SZ, UK

**Keywords:** classification algorithms, essential tremor, personalized medicine, handwriting analysis, linear discriminant analysis, machine learning, principal component analysis, support vector machines

## Abstract

Background: Essential tremor (ET) is among the most common neurological disorders, requiring precise diagnosis and severity assessment for personalized and effective management. Methods: This study explores an innovative approach to evaluate ET severity using the gold-standard Archimedes spiral test. The family-based dataset covers the entire range of tremor severity, from very mild (level 1) to advanced stages, offering a valuable resource for studying early diagnosis and tracking disease progression. The proposed method introduces a machine learning pipeline that combines Principal Component Analysis (PCA), linear discriminant analysis (LDA), and support vector machines (SVMs) to classify ET severity based on Archimedean spiral radius data. Results: By incorporating the Fahn–Tolosa–Marin Tremor Rating Scale (FMT-TRS), the pipeline effectively distinguishes between tremor presence and severity. Its robustness was demonstrated through rigorous cross-validation and tests involving Gaussian noise perturbations. Conclusions: These results underscore the machine learning-based pipeline’s potential as a non-invasive and trustworthy diagnostic tool for clinical use and telemedicine applications. Moreover, the combination of geometric features, FMT-TRS scores, clinically oriented evaluation metrics, and classical statistical and machine learning models offers a robust, interpretable, explainable, and clinically meaningful analytical framework.

## 1. Introduction

Essential tremor (ET) affects millions of individuals globally, often leading to significant motor disability and reduced quality of life. Recent prevalence studies suggest that ET is one of the most common movement disorders, with increasing recognition of its heterogeneity and impact [1]. The Movement Disorder Society defines essential tremor (ET) as an isolated tremor syndrome characterized by a bilateral upper-limb action tremor with a minimum duration of three years [2,3], which may or may not be accompanied by tremor in other regions, such as the head, voice [4], or lower limbs, and with no other neurological impairments, as seen in Parkinson’s disease [5]. The disease can become highly disabling and, in familial cases [6], the tremor may appear at very early ages. For this reason, early diagnosis of this neurodegenerative disorder is essential, as it can significantly improve patients’ overall well-being and enhance the effectiveness of available treatments. Despite its prevalence, diagnosing ET remains challenging due to its overlap with other movement disorders, such as Parkinson’s disease (PD), and the subjective nature of traditional clinical evaluations [7,8]. Advancements in neuroimaging, such as functional MRI (fMRI) and Diffusion Tensor Imaging (DTI), have provided valuable insights into the neurological underpinnings of ET [9,10]. These tools have been instrumental in exploring brain regions associated with motor control abnormalities in ET patients. However, high costs, limited accessibility, and the complexity of interpreting results make them impractical for routine clinical management in primary care and monitoring.

Action tremor significantly impairs motor function and is typically observed within the 4–12 Hz range [3]. It can be measured with high precision with handwriting analysis, particularly using the Archimedes spiral test, which is a robust clinical assessment and the gold standard for ET diagnosis [11,12,13]. The affordable, simple, and non-invasive method has been adapted for telemedicine using digital tools. Spiral drawing can be captured across a range of devices, from high-precision tablets, such as Wacom, to widely accessible consumer devices, including smartphones and standard tablets, without requiring specialized infrastructure. Digitally acquired spirals can be represented as spatial trajectories (time series), enabling precise computation of their dynamics and tremor-related metrics, such as amplitude, velocity, and frequency over time [13]. These systems capture not only geometric features (x, y coordinates, azimuth angle) but also pressure data and, in advanced configurations, in-air trajectories (Wacom, Europe). Recording movement both on and above the surface enhances the assessment of fine motor control and brain performance, particularly during initial movement phases when disease-related abnormalities are most evident [14].

The latest developments integrate advanced signal-processing and machine learning techniques to extract relevant information from spiral drawings, including both image and time-series data [13,15]. Additionally, nonlinear biomarkers, such as entropy and fractal dimension, in combination with traditional kinematic features, have shown strong potential for distinguishing ET patients from healthy controls [16]. In the last decade, methodological evolution was considerable: early studies were based on digital tablets connected to a computer that captured x-y and pressure data, whereas modern approaches employ smartphones, stylus-based tablets, or scanned paper spirals with machine vision algorithms for automated classification [17]. Extracted metrics include basic kinematics (amplitude, speed, acceleration) and nonlinear features (entropy, fractal dimension), among others, and usually integrate classifiers such as SVMs, CNNs, or LSTMs [16,18,19]. Undoubtedly, digital spiral assessment improves the differentiation between essential tremor alterations and stress-related drawing irregularities by providing objective, high-precision measurements that go beyond subjective visual ratings [20]. Consequently, tremor quantification becomes more sensitive and reproducible, supporting diagnosis, longitudinal monitoring, and intervention studies. Automated tremor amplitude computation from digitized spirals correlates strongly with manual measurements and clinician ratings, supporting the use of digital spiral metrics as trustworthy biomarkers in ET clinical issues [21,22].

Looking ahead, integrating digital spiral assessments into telemedicine and home-based monitoring frameworks represents a promising approach and a challenge. Indeed, these methodologies are progressively establishing themselves as robust tools for early diagnosis, disease progression monitoring, and therapeutic response assessment, extending their utility beyond conventional clinical settings [13,23,24]. Recent advancements have enabled the development of non-invasive machine learning pipelines for essential tremor (ET) classification, leveraging data from sources such as accelerometry and digital spiral tests [25,26,27]. These approaches have demonstrated promise in distinguishing ET from other movement disorders, including Parkinson’s disease, and in quantifying tremor severity [17,24]. Digital spiral assessments, in particular, show high consistency and strong correlations with clinical tremor rating scales, such as TETRAS, which is mainly used to evaluate severe tremor and measure kinetic tremor amplitude [22]. Despite these advances, integrating clinical severity scales that cover the full tremor spectrum, such as the Fahn–Tolosa–Marin Tremor Rating Scale (TRS) [28,29] into automated pipelines remains limited. These scales represent an opportunity to enhance both diagnostic [29] accuracy and interpretability [29,30,31]. In fact, even though the TRS is widely recognized for its reliability in evaluating tremor severity [8,32], its application in Artificial Intelligence (AI)-based diagnostic frameworks is an emerging challenge.

This study proposes a robust machine learning-based diagnostic pipeline that leverages features derived from Archimedean spiral tests and integrates the Fahn–Tolosa–Marin Tremor Rating Scale (FTM-TRS) to classify tremor severity levels. It included families with identified genetic loci affected by essential tremor (ET), with participants drawn from a previous familial ET study and control individuals. Family-based recruitment captured the full spectrum of tremor severity, from very mild (level 1) to advanced stages, providing a unique dataset for researching early diagnosis and disease progression. In contrast, most existing datasets are derived from patients presenting moderate to severe tremor, limiting their utility for studying early-stage manifestations. Data acquisition was performed using a Wacom digitizing tablet. By combining dimensionality reduction techniques (PCA, LDA) with supervised classification (SVM), we achieved high accuracy and robustness, validated with rigorous statistical testing and cross-validation.

Our approach bridges the gap between traditional diagnostic methods and modern machine learning, offering a non-invasive, scalable solution for ET assessment. Additionally, clinically interpretable TRS scores, combined with geometric features, such as spiral radius and signal residues, enable intuitive analysis of motor performance. Dimensionality-reduction methods (e.g., PCA, LDA) simplify high-dimensional data into clinically meaningful components, while SVMs capture complex patterns without losing interpretability, supporting the use of digital motor biomarkers in clinical decision-making.

## 2. Materials and Methods

Around fifty patients with ET and fifty unaffected controls were recruited from a descriptive study of familial and sporadic ET cases in the Movement Disorders Unit at the Donostia University Hospital, San Sebastian, Spain, from January 2015 to June 2017. All participants were diagnosed by a movement disorder specialist using established clinical criteria. The exclusion criteria for ET participants included isolated moderate-to-severe head tremor, Parkinsonism, ataxia, orthostatic hypotension, gaze palsy, and a secondary cause of tremor. Healthy controls were excluded if they had any neurologic illness or family history of ET after clinical evaluation and medical record review [6,8,16]. Tremor severity in the ET participants was evaluated using the Fahn–Tolosa–Marin Tremor Rating Scale (TRS) [28].

All participants were provided with a detailed explanation of this study and gave written and verbal informed consent before any procedures were conducted. This study was approved by the Ethics Committee of Donostia Hospital (Comité Ético de Investigación Clínica del Área Sanitaria de Gipuzkoa, Acta nº 4/2010, Paseo Doctor Begiristain, 109, San Sebastián, 20014, Spain). The current investigation used the BIODARW database to conduct the experimentation. This dataset, as referenced in previous works [14,33], is specifically designed to analyze and diagnose movement disorders using handwriting analysis, focusing on the Archimedes spiral test. The dataset aims to enhance the understanding of fine motor skills, particularly in the context of essential tremor (ET). The dataset integrates neuroimaging and handwriting analysis as a broader initiative to improve early diagnostic approaches for ET. The participants in this study were classified into two groups: ET patients and controls. These individuals were recruited from pre-existing patient and control cohorts. Among the various activities performed, the participants were asked to draw Archimedes spirals with both hands using a digitizing tablet to capture the samples. This version of the BIODARW dataset comprises 53 samples, including 24 from the ET group and 29 from the control group, facilitating a comparative analysis of current findings with those from previously reported studies.

### 2.1. Data Acquisition

The participants were asked to draw an Archimedean spiral with both hands using a Wacom 4 digitizing tablet. Accurate capture and analysis of tremor dynamics (4–12 Hz) requires selecting an appropriate sampling rate for signal acquisition. In this study, a sampling frequency of 200 samples per second was employed. This rate provides sufficient temporal resolution to fully resolve ET oscillatory patterns while ensuring the tremor waveform is reproducible across repeated measurements. Each sample is stored in matrix format, where columns contain x and y coordinates, timestamp, pen-up values (1 when the pen touches the paper), azimuth, altitude angle, and pressure. This high temporal resolution enables capturing fine-grained temporal variations in movement, which is essential for diagnosing tremor disorders, such as ET (Figure 1).

The visual differences underscore the significance of spatial coordinates (x, y) as features for tremor detection and classification. We worked mainly with X and Y coordinates because any digital tablet can obtain these parameters. Therefore, the cost of implementing and industrializing the gathering process will be drastically reduced. Additionally, X and Y coordinates provide enough information to diagnose ET [14,18].

### 2.2. Demographics and Tremor Severity Distribution

The dataset comprised Archimedes spiral handwriting samples alongside associated metadata, including tremor presence and severity levels. The tremor severity levels were stratified into three groups based on the TRS: (1) CoG (Control Group), including subjects without tremor; (2) LtG (Low Tremor Group), including subjects exhibiting mild tremor; and (3) HtG (High Tremor Group), including subjects with severe tremor (Figure 2).

The BIODARW dataset’s demographic and tremor severity distributions are illustrated in Figure 3, which shows the following:Tremor Presence: Out of 53 spirals, 24 exhibited tremors (tremor-positive cases), while 29 were tremor-free (controls).Gender Distribution: The dataset comprised more female samples than male samples, with 28 females and 25 males.Age Distribution: The participants ranged in age from 10 to 80, with a peak distribution between 50 and 70, aligning with the typical ET onset.TRS Scores and Tremor Levels: The TRS scores exhibit a highly skewed distribution, with most subjects having low scores indicative of mild or no tremor. The tremor levels were stratified into three categories: control (level 0), low tremor (level 1), and high tremor (level 2).

### 2.3. Correlation Analysis of Key Features

A correlation matrix was computed to analyze relationships between key variables such as TRS score, age, and tremor level. The strong correlation between the TRS and tremor level (0.87) confirms the suitability of the TRS-derived tremor levels for capturing severity and classification tasks. The moderate correlations between the TRS and age (0.62) and age and tremor level (0.74) indicate an age-related trend in tremor severity, highlighting age as a potential confounder (Figure 3).

These findings validate the dataset’s relevance for machine learning and its robust feature extraction and classification potential.

### 2.4. Data Preprocessing

The initial preprocessing step involved filtering the raw dataset. The data consisted of hand-drawn Archimedes’ spirals, including variables such as x-coordinates, y-coordinates, pen pressure, and pen-up information. Filtering was applied to remove irrelevant or noisy data points, thereby preserving the most informative signals for subsequent analysis. This step ensures data integrity for robust feature extraction and classification.

The filtered spiral data was resampled to a fixed length of 4096 points to align the input data for consistent analysis. This resampling was critical to address variations in the number of recorded points per subject. Interpolating the x, y, pen-up, and pressure components ensured compatibility with machine learning models by preserving spiral data fidelity (Figure 4).

### 2.5. Feature Extraction

The raw data underwent a series of preprocessing steps to enhance signal quality and extract relevant features. The radius variable was computed by transforming the Cartesian coordinates (x, y), in each sample n ($x_{n},y_{n}$), obtained from the Archimedes spiral into polar coordinates using the following Formula (1):
(1)$$r_{n}=\sqrt{x_{n}^{2}+y_{n}^{2}}.$$

This transformation facilitated the analysis of global spatial variability within the spiral trajectories. By reducing the spatial movement into a single variable, the radius provided a compact yet comprehensive representation of the drawing patterns (Figure 5).

The radius was essential for capturing broad-scale tremor dynamics. It allowed us to assess consistency or irregularity in the drawing process, particularly highlighting oscillatory patterns and deviations characteristic of ET. These patterns were especially significant in distinguishing between severity levels stratified by the TRS.

A discrete cosine transform (DCT) and its residues, as described by [34], were applied in the feature extraction pipeline to analyze the frequency domain characteristics of the data. DCT is particularly effective in compacting signal energy into a few coefficients, facilitating the representation of complex motor control patterns observed in spiral drawings.

We computed the radius time-series correlation matrices for both cases to further investigate the tremor dynamics in the ET and control groups. The visualizations below provide a comprehensive view of temporal correlations across radius values (Figure 6 and Figure 7).

#### 2.5.1. Preprocessing

A smoothing operation was applied during preprocessing to enhance the stability and reliability of the radius time series. The raw radius values were smoothed using a rolling mean with a window size defined as round (l/denom), representing the length of the time series. Denom is a divisor chosen to balance noise reduction with meaningful variability retention. This smoothing step helped mitigate abrupt fluctuations in the raw data while preserving essential trends, ensuring the data’s suitability for further analysis.

DCT residues and radius features from the Cartesian coordinates of Archimedes spirals were used in preprocessing. These residues were derived from a discrete cosine transformation of the x and y coordinate time series and highlight frequency variations sensitive to ET. After smoothing with a rolling mean for signal stability, normalization was applied to align the scale of DCT residues with the radius features. This ensured that all features (radius and DCT residues) were on the same scale, enabling joint analysis.

The impact of this preprocessing step is evident in the correlation matrices generated for the control and ET groups, as shown in Figure 8. For the control group, the smoothing process resulted in highly consistent and uniform correlation patterns, reflecting the natural stability of motor control. Conversely, for the ET group, the correlation matrix revealed disrupted patterns and increased variability, aligning with the irregularities introduced by the tremor. These matrices provide a strong foundation for extracting correlation-based features, underscoring the importance of preprocessing in highlighting group differences critical for machine learning classification.

#### 2.5.2. Dimensionality Reduction in the Pipeline

The pipeline integrates Principal Component Analysis (PCA) to reduce the high dimensionality of spiral test data (4096 features) while retaining 95% of the variance [35,36]. Linear discriminant analysis (LDA) further optimizes class separability by projecting data onto a lower-dimensional space. Two discriminants (LDA1 and LDA2) were extracted for classification and visualization, enhancing interpretability [27,37]. Linear discriminant analysis (LDA) was utilized as a dimensionality reduction technique and classifier. LDA projects data onto axes that maximize class separability, enabling dimensionality reduction and interpretable feature extraction that highlight group differences. This approach was particularly valuable for analyzing tremor severity levels, as the LDA-transformed data emphasized distinctions between control subjects and different tremor severity groups. The LDA model was configured to use two components, facilitating the visualization of class separability while retaining sufficient information for classification tasks. LDA was employed to identify and quantify the contribution of features to class separability. LDA creates linear discriminants (LDA1 and LDA2) that maximize class separability. To analyze the contribution of features, we computed the magnitude of each feature’s impact on LDA1 and LDA2. For the first discriminant axis, LDA1, Feature 1 contributed significantly, while Feature 2 had no measurable influence. Conversely, for LDA2, Feature 2 dominated, with no quantifiable contribution from Feature 1. This indicates that Feature 1 and Feature 2 are orthogonal in their discriminative ability, each specializing in class separation along distinct axes.

The integration of PCA and LDA was motivated by several factors:Dimensionality Reduction: The high dimensionality of time series (e.g., radius and DCT residue time series) poses computational challenges and increases the risk of overfitting. PCA effectively reduced dimensionality by retaining components that preserved most of the variance in the dataset.Noise Reduction: PCA filtered out noise from redundant and irrelevant dimensions by projecting the data into a lower-dimensional subspace. This preprocessing step ensured that LDA could focus on the most informative features for class separation.Numerical Stability: LDA’s requirement to invert a covariance matrix becomes unstable in high-dimensional settings. PCA mitigated this by reducing the feature space.Discriminative Power: While PCA prioritizes variance preservation, it does not account for class separability. LDA complemented PCA by projecting the reduced components into a space that maximized between-class variance while minimizing within-class variance.

This approach aligns with recommendations in the machine learning literature [38,39] where PCA is often used to enhance the performance and computational efficiency of LDA. Similarly, refs. [14,17,34,40] demonstrated the utility of dimensionality reduction techniques for classifying tremor data, validating the pipeline’s relevance to essential tremor studies.

The experiment’s effectiveness was extended to features derived from discrete cosine transform (DCT) residues. DCT residues provided compact representations of frequency domain characteristics, reflecting localized deviations in tremor signals. This integration of DCT residues into the PCA → LDA pipeline further enhanced class separability by leveraging orthogonal information compared to global radius features. After smoothing with a rolling mean for signal stability, PCA was used to reduce the dimensionality of the DCT residues, preserving the most informative components. LDA was then used to project the reduced features into a space that optimized class separability.

The TRS scores were integrated as numerical and categorical features to enhance model interpretability and classification accuracy. TRS scores provide a clinically validated measure of tremor severity [29,33]. Previous studies have shown that including structured clinical scores, such as the TRS, improves the diagnostic performance of machine learning pipelines for tremor classification [41,42].

### 2.6. Classification Pipeline

Multiple machine learning algorithms, including support vector machine (SVM), k-nearest neighbor (k-NN), linear discriminant analysis (LDA), and Random Forest, were employed to evaluate the robustness of the classification pipeline. This comparative analysis assessed model performance on the tremor severity classification task [27,32,36]. Diverse classifiers were used to balance sensitivity, specificity, and robustness, following best practices in tremor classification research [19,27], as listed in Table 1.

Implementation Details

PCA: The PCA step reduced the original 4096 features to a manageable number of components while retaining at least 95% of the variance. The number of retained components was determined empirically using cumulative variance analysis.LDA: The LDA model was applied to the reduced components to enhance class separability. Two linear discriminants (LDA1 and LDA2) were extracted, providing a two-dimensional representation of the data for visualization and classification.Classifier Integration: LDA-transformed features were subsequently used as input to the classifiers.

The hyperparameters for the SVM, k-NN, and Random Forest classifiers were optimized based on established methodologies for nonlinear separability and generalization [32]. Specifically, the SVM employed an RBF kernel with C = 0.1–1 and gamma = 0.1, ensuring a smooth decision boundary while minimizing overfitting. The k-NN classifier, configured with k = 5–10 and a Euclidean distance metric, balanced sensitivity and specificity. The Random Forest model, consisting of *n* = 100 decision trees, utilized random feature selection and hyperparameter tuning through grid search, enabling it to capture nonlinear patterns effectively.

All classifiers were evaluated using the PCA-transformed and LDA-reduced features to ensure comparability. The classifier was trained and tested using a Leave-One-Out cross-validation (LOOCV) strategy to maximize the use of the limited dataset and ensure robust evaluation [43,44,45]. Additionally, performance was validated using accuracy, precision, recall, and F1-score metrics, with a confusion matrix to assess classification consistency across tremor severity levels.

To evaluate the robustness of the classification pipeline, Gaussian noise was added to the LDA-reduced feature space $X_{lda_{noisy}}(2)$:
(2)$$X_{lda_{noisy}}=\;X_{lda+}N(0,\;0.01),$$
where N(0,0.01) represents Gaussian noise with a mean of 0 and a standard deviation of 0.01. This simulated a small random perturbation in the input data. The model’s performance was assessed using 5-fold cross-validation with the noisy data. Stratified K-Fold cross-validation (5 splits) was employed to ensure balanced class distributions across folds. This approach divides the dataset while maintaining class balance within each fold, allowing for a fair evaluation of model performance on imbalanced datasets [45,46].

### 2.7. Statistical Validation

Correlation matrices were computed for both the ET and control groups using Pearson correlation coefficients to evaluate the variability in and consistency of the radius time-series data. The diagonal elements were excluded from the analysis to focus on off-diagonal correlations, representing relationships between non-consecutive time points.

Summary Statistics: The mean and standard deviation of off-diagonal correlation values were calculated for each group to quantify overall consistency and variability.

Mann–Whitney U Test: A non-parametric Mann–Whitney U test was performed to evaluate the statistical significance of differences in off-diagonal correlation values between the ET and control groups. The test was chosen due to its robustness in handling non-normal distributions.

Visualizations were created to analyze these differences further:Histograms were used to display the distribution of correlation values for the ET and control groups.Boxplots were used to highlight the groups’ medians, interquartile ranges, and variability differences.

## 3. Results

### 3.1. Correlation Matrix

For the ET cases, the correlation matrix in Figure 7 exhibits a fragmented structure with lower overall correlations (minimum value: 0.4381). These patterns align with tremor-induced irregularities characterized by less smoothness and higher variability in radius values. Conversely, the control matrix demonstrates a smoother and more consistent correlation pattern (minimum value: 0.5518), reflecting stable and predictable movements in the control subjects.

Figure 8 shows the correlation matrices for DCT residues, which further validate the distinction between the ET and control cases. The matrix exhibits a fragmented structure for the ET cases with lower overall correlations (minimum value: 0.6839). This fragmented pattern aligns with the irregularities caused by tremor-induced variability in motor control, as captured in the frequency domain characteristics by DCT residues. In contrast, the control correlation matrix displays a smoother, more uniform pattern with higher correlations (minimum value: 0.7737). This reflects healthy individuals’ stable and predictable motor control, as captured by the DCT residue analysis.

These quantifiable differences in correlation values observed consistently across both the spatial (radius) and frequency (DCT residues) domains underscore the complementary nature of these features in characterizing tremor severity. This reinforces the utility of correlation-based metrics derived from multiple domains as robust features for machine learning classification.

### 3.2. Statistical Analysis

#### 3.2.1. Radius

To analyze the consistency of the radius time-series data across the ET and control groups, we computed the average correlation and its standard deviation for each group, excluding diagonal elements of the correlation matrices.

Control Group:The average correlation of 0.921 indicates highly consistent and similar radius patterns among the control subjects.The standard deviation of 0.061 indicates low variability, reinforcing the uniformity of the radius data in the control group.

ET Group:The average correlation of 0.853 was lower than that of the control group, reflecting disruptions and higher variability in the radius time series caused by the tremor.The standard deviation of 0.077 suggests higher variability than the control group, indicating irregular patterns in the ET group’s radius data.

The Mann–Whitney U test confirmed the statistical significance of these differences, yielding a *p*-value of 0.0001 and providing strong evidence against the null hypothesis. These results support the hypothesis that essential tremor disrupts natural movement consistency, as reflected in the lower and more variable correlations observed in the ET group (Figure 8).

To evaluate the variability in and consistency of the radius time-series data across the ET and control groups, visualizations, including the histograms and boxplots in Figure 9, were created to compare their correlation distributions.

Histogram Analysis:The control group exhibits higher correlation values, peaking near 1.0, indicating consistent and uniform radius patterns.The ET group displays a broader distribution, with correlation values across a wider range and many falling below 0.9, reflecting increased variability caused by tremor-induced irregularities.While the distributions overlap, the distinct patterns highlight group differences.

Boxplot Analysis:The median correlation for the control group is visibly higher, close to 0.95, reflecting greater consistency in radius data.The ET group shows a lower median correlation (~0.88) and a more extensive interquartile range (IQR), indicating higher variability in the radius time series.

#### 3.2.2. DCT Residues

To analyze the consistency of the DCT residues across the ET (essential tremor) and control groups, we computed the average correlation and its standard deviation for each group, excluding the diagonal elements of the correlation matrices.

Control Group:The average correlation of 0.996 reflects the high consistency and smooth patterns in the frequency domain representation of the control subjects’ movement.The standard deviation of 0.008 indicates extremely low variability, underscoring the uniformity of the DCT residues in the control group.

ET Group:The average correlation of 0.9754 is lower than that of the control group, highlighting disruptions in the frequency domain due to tremor-induced irregularities.The standard deviation of 0.0230 suggests higher variability compared to the control group, indicating less uniformity in the frequency domain representation of the ET subjects.

The Mann–Whitney U test confirmed the statistical significance of these differences, yielding a *p*-value of 0.001 and providing strong evidence against the null hypothesis. These results demonstrate that essential tremor disrupts natural frequency domain consistency, as reflected in the lower and more variable correlations observed in the ET group. These findings align with the hypothesis that tremors affect motor control consistency across the spatial (radius) and frequency (DCT residues) domains, emphasizing the potential of DCT-derived features for distinguishing ET subjects from controls in machine learning classification tasks.

Visualizations, including histograms and boxplots (Figure 8), were created to evaluate the variability in and consistency of DCT residue data across the ET and control groups. These visualizations provided additional insights (Figure 9):

Histogram Analysis:The control group exhibits higher correlation values, tightly clustered near 1.0, indicating consistent and uniform frequency domain patterns.The ET group shows a broader distribution, with correlation values across a wider range and many falling below 0.99, reflecting tremor-induced irregularities in frequency domain data.

Boxplot Analysis:The median correlation for the control group is nearly 1.0, emphasizing the high consistency of DCT residues in this group.The ET group has a visibly lower median correlation (~0.98) and a more extensive interquartile range (IQR), signifying higher variability caused by tremors.

### 3.3. Dimensionality Reduction

The LDA of feature contributions revealed that Features 1 and 2 play distinct and complementary roles in class separability. For LDA1, Feature 1 contributed significantly, demonstrating its critical role in defining the first discriminant axis. In contrast, Feature 2 had no measurable influence on LDA1. For LDA2, Feature 2 emerged as the dominant contributor, while Feature 1 had no measurable impact. These results indicate that Features 1 and 2 are orthogonal in their discriminative properties, each specializing in class separation along a specific axis (Figure 10).

Visualization of the feature contributions to LDA1 and LDA2 (Figure 11) highlights this orthogonality, with Feature 1 dominating the first axis and Feature 2 dominating the second. These findings demonstrate the effectiveness of LDA in creating discriminative features that are orthogonal and uncorrelated (Figure 11).

The LDA-transformed features (shown in Figure 12) lead to better classification performance than PCA alone (as depicted in the PCA chart). The dimensionality reduction pipeline combining PCA and LDA effectively extracted features that optimized class separation. Classifiers trained on these LDA-transformed features exhibited high classification accuracy, confirming the robustness and utility of the extracted features for machine learning applications. These results validate the potential of using a PCA-LDA pipeline for feature extraction and dimensionality reduction in tasks requiring high-class separability.

### 3.4. Classification

Multiple machine learning models were evaluated using Leave-One-Out validation and cross-validation techniques to assess the classification pipeline’s performance and robustness. All classifiers were optimized to achieve better generalization on the test dataset, according to the configuration defined in Table 1. Table 2 shows all the metrics for model performance comparison using the Leave-One-Out validation for radius. SVM with an RBF kernel and the Random Forest model achieved a training accuracy of 100% and a test accuracy of 96.23% and 94.34%, respectively, capturing nonlinear patterns in the LDA-transformed feature space, which suggests some degree of overfitting in this dataset in training, but with optimum results for the test in all the metrics. The k-NN classifier obtained accuracies of 95.94% for training and 92.45 for testing, with lower recall (86.67%) and F1-score (89.58) values, suggesting the worst performance. Finally, the LDA classifier obtained accuracies of 94.12% for training and 90.57% for testing, with the lowest scores for precision (89.56%), recall (84.87%), and F1-score (89.58%), suggesting underperformance due to the dataset size.

The same machine learning models were evaluated to assess the classification pipeline’s performance and robustness using DCT residues as features. All classifiers were optimized to achieve better generalization on the test dataset, according to the configuration defined in Table 1. The evaluation utilized Leave-One-Out validation and cross-validation techniques, as previously described for the radius features. For the DCT residues, SVM with an RBF kernel and the Random Forest model achieved training accuracies of 100% and test accuracies of 98.11% and 94.34%, respectively, improving on the previous results. These models successfully captured nonlinear patterns in the LDA-transformed feature space, suggesting potential overfitting during training. However, they achieved optimal performance on all test metrics (Table 3). The k-NN classifier reached a training accuracy of 95.94% and a test accuracy of 92.45%, showing improved test performance compared to the previous results, with a precision of 96.08%, a recall of 86.67%, and an F1-score of 89.58%. Finally, the LDA classifier obtained a training accuracy of 94.12% and a test accuracy of 90.57%, but it showed the lowest scores among the test metrics, with a precision of 89.56%, a recall of 84.87%, and an F1-score of 86.86%, suggesting that the limited dataset size may have contributed to its underperformance.

These results indicate the pipeline’s ability to adapt to different feature types, with DCT residues complementing the radius results in capturing class distinctions. SVM demonstrated robust performance across both feature types, further validating its utility for tremor classification. Finally, Figure 12 illustrates the accuracy for the failed subjects with Leave-One-Out validation for all the classifiers and train vs. test. This provides a helpful tool for evaluating and interpreting subject-specific outcomes and analysis.

LDA demonstrated its value in dimensionality reduction and classification by projecting the data to maximize class separability. Our two-component LDA model allowed clear distinctions between the control subjects and the tremor severity groups, aiding visualization and classification of motor control variations. These results confirm the strong potential of the pipeline for clinical use, with SVM and LDA proving the most effective for essential tremor classification. The performance of the PCA → LDA → SVM pipeline was rigorously evaluated using multiple validation approaches, including hold-out validation, cross-validation, stratified K-Fold validation, and robustness testing under Gaussian noise perturbations. The results indicate exceptional classification accuracy and robustness across both radius and DCT residue features, demonstrating the pipeline’s ability to effectively generalize and retain discriminative power under challenging conditions Table 4.

SVM with an RBF kernel, evaluated using the hold-out method, achieved optimal training results for both radius and residues across all metrics. However, its test performance was lower, with radius achieving an accuracy of 81.82% and the poorest performance on the selected metrics, namely, a precision of 53.33%, a recall of 66.67%, and an F1-score of 58.33%, demonstrating limited generalization capability. Using residues, the model showed improved performance on the test set, although some metrics did not reach optimal values. In contrast, cross-validation with a 5-fold setup further validated the pipeline, yielding a perfect mean cross-validation score of 100% for training across all metrics and optimal results on the test set. Stratified K-Fold cross-validation, designed to maintain balanced class distributions across folds, corroborated these findings with a mean score of 100%. For DCT residues, the pipeline similarly achieved perfect scores of 100% for both the train and test sets, reflecting the high discriminative capability of DCT-derived features. Gaussian noise was added to the LDA-transformed data to assess robustness under perturbations. Even under these conditions, the model maintained perfect classification performance, with a mean cross-validation score of 100% and a mean stratified CV score of 100%. These results confirm that the pipeline is resilient to noise and preserves its discriminative power under challenging scenarios. All metrics indicate efficient classification performance, validating the pipeline’s robustness and effectiveness for tremor severity management.

## 4. Discussion

### 4.1. Interpretation of Results

The significant statistical differences in correlation patterns between the ET and control groups, as evidenced by a highly significant *p*-value (0.001), underscore the disrupted and variable nature of radius time series in the ET subjects compared to the smooth and consistent patterns observed in the controls. These findings align with the hypothesis that essential tremor disrupts the natural consistency of movement, introducing irregularities in motor control that are reflected in hand-drawn spirals [7,29,33]. The lower average correlation (0.853) and higher variability in the ET group compared to the control group (0.921) highlight the fragmented and inconsistent patterns caused by tremors. These results underscore the utility of off-diagonal correlation metrics as features for classification tasks, providing a robust and quantitative foundation for distinguishing between tremor severity and standard motor control. The findings validate the potential of radius time-series features for machine learning applications, further supporting their relevance in developing non-invasive diagnostic tools for ET and its severity.

While radius features provided interpretable insights into the global movement patterns of tremor patients, DCT residues captured local, frequency domain irregularities that are less evident in raw radius data. Both features demonstrated excellent classification performance even before dimensionality reduction, with the SVM model achieving close test accuracy (radius: 96.23%, DCT residues: 98.11%). This suggests that while these features are distinct, they are equally effective in distinguishing tremor severity and control subjects. LDA, applied after PCA, maximizes class separability by projecting data onto axes that emphasize inter-class differences while minimizing intra-class variance. LDA creates a feature space where severity classes are well-separated. This is particularly critical for ET classification, where subtle differences in movement patterns require precise feature extraction. The results showed that the two LDA components provided distinct and orthogonal features, further simplifying the classification task.

The results demonstrate that the PCA → LDA → SVM pipeline is highly robust and generalizable, regardless of whether radius features or DCT residues are used. The consistently high performance of the SVM model across both feature types highlights its adaptability to different data characteristics. The nonlinear capabilities of the RBF kernel in SVM enabled it to capture complex patterns in the DCT-transformed feature space, achieving the highest test accuracies among all models for both feature sets. The combination of PCA’s variance preservation, LDA’s discriminative feature extraction, and SVM’s capacity to model complex relationships ensures the pipeline’s robustness and high accuracy. Additionally, SVM’s hyperparameters (C and gamma) were optimized to balance complexity and generalization, enhancing performance.

This integrated pipeline was validated across multiple evaluation methods, consistently achieving perfect classification scores. The robustness of Gaussian noise perturbations highlights its potential applicability to real-world, noisy datasets. These findings confirm the suitability of PCA, LDA, and SVM as an effective combination for non-invasive diagnosis of ET severity.

### 4.2. Comparison with Existing Work

This study builds on and complements recent advancements in machine learning applications for essential tremor (ET) classification by integrating clinically validated severity scales (FTM-TRS) with robust spatial (radius) and frequency domain (DCT residues) features. While prior research has focused on single-domain features or sensor-based metrics, this work uniquely combines these approaches with advanced dimensionality reduction and classification techniques, offering a comprehensive and scalable solution for tremor classification.

As part of ongoing research in our group, ref. [34] explored DCT residues for analyzing tremor signals but focused primarily on frequency domain characteristics. While effective in identifying localized motor control disruptions, this approach did not consider integrating spatial features like radius. Our study extends this work by combining DCT residues with radius features, demonstrating that both domains provide complementary insights for essential tremor (ET) classification. This combined feature set enhances robustness and generalizability, as evidenced by consistently high performance across validation techniques. Similarly, ref. [27] developed accelerometry-based methods to quantify tremor severity using FTM criteria but did not incorporate advanced dimensionality reduction or classification techniques. Their work, while clinically relevant, remained limited to translating clinical scales to sensor-based methods, lacking the scalability of automated pipelines. In contrast, our approach demonstrates that integrating FTM-TRS into machine learning models can bridge the gap between clinical evaluations and objective, automated tremor diagnostics.

Recent reviews, such as those by [7,47], emphasize the growing need for objective and automated methods in tremor assessment. Our pipeline addresses these challenges by achieving high accuracy and robustness while leveraging low-cost, non-invasive handwriting data. This aligns with the demand for scalable, telemedicine-compatible solutions that are accessible to diverse clinical environments. Similarly, our group has also leveraged advanced deep learning techniques, as demonstrated in [18], using Empirical Mode Decomposition (EMD) and Long Short-Term Memory (LSTM) networks to process handwriting data for ET classification. This approach highlights the potential of deep learning methods to handle raw, unprocessed data directly, in contrast to the preprocessing steps required in this study. However, while LSTM networks excel at capturing temporal dependencies, they often require large datasets to achieve high accuracy. In contrast, the PCA → LDA → SVM pipeline demonstrated high robustness and accuracy on a smaller dataset, making it more immediately applicable in clinical contexts with limited data availability. Ref. [3] utilized advanced neuroimaging metrics combined with machine learning to identify ET. While their work offers valuable insights into brain network disruptions in ET patients, it requires costly imaging equipment, making it less accessible for routine diagnostics. In contrast, the proposed pipeline focuses on low-cost, non-invasive handwriting data, providing a practical alternative for wider clinical adoption and telemedicine applications. Our group is also exploring entropy-based methods, which have been shown to effectively quantify variability and irregularity in tremor signals [5]. This study emphasized the diagnostic relevance of entropy features but did not integrate clinically validated scales like FTM-TRS into the analysis. By incorporating both FTM-TRS and complementary radius and DCT residue features, our current study enhances interpretability and clinical relevance, bridging the gap between purely mathematical features and clinically actionable outcomes.

Lastly, an ongoing study by our group [33] combines neuroimaging metrics with handwriting features to advance ET diagnosis and management. While neuroimaging provides insights into structural and functional disruptions, it is resource-intensive and less accessible for routine diagnostics. The current work complements this approach by focusing on handwriting data alone, demonstrating that inexpensive, non-invasive methods can achieve high accuracy. Additionally, integrating DCT residues into our pipeline highlights the potential of frequency domain features, which may serve as surrogates for neuroimaging biomarkers in specific contexts.

This study bridges the gap between these approaches by achieving high accuracy with interpretable machine learning models while maintaining computational efficiency. Additionally, the pipeline’s robustness was validated with Gaussian noise perturbations and multiple validation methods, further emphasizing its suitability for real-world applications. The inclusion of Gaussian noise testing further distinguishes the pipeline, demonstrating robustness against input perturbations, an aspect often overlooked in prior research. These findings underscore the practical advantages of the proposed method for real-world clinical applications, including telemedicine and industrial deployment.

### 4.3. Limitations

While this study demonstrated the utility of radius and DCT residue features in classifying essential tremor (ET) severity, several limitations warrant consideration. First, the dataset size is relatively small, which limits the generalizability of our findings across diverse populations and tremor subtypes. Second, while DCT residues provided valuable insights into local, frequency domain irregularities, other potentially informative features, such as azimuth and pressure signals, were not included in this study. Omitting these features might limit the full diagnostic potential of the classification pipeline. Third, the preprocessing steps, including rolling-mean smoothing and normalization, add complexity to the pipeline, potentially increasing implementation time and costs in clinical or telemedicine settings. While the PCA → LDA → SVM pipeline proved robust and effective, it requires hand-tuned dimensionality reduction steps, which could be replaced by end-to-end deep learning models in future studies. These models may eliminate the need for extensive preprocessing, reducing computational overhead and simplifying industrial deployment. Lastly, while robustness to Gaussian noise was validated, further testing under real-world noisy conditions, such as varying handwriting styles or environmental interference, is necessary to ensure reliability in clinical applications. Future work should address these limitations by leveraging larger, more diverse datasets, integrating additional features, and exploring deep learning models to streamline preprocessing and improve scalability.

### 4.4. Broader Implications

The integration of the PCA → LDA → SVM pipeline with clinical tools such as the Fahn–Tolosa–Marin Tremor Rating Scale (FTM-TRS) represents a novel approach for classifying essential tremor (ET) severity and offers significant potential for clinical practice. The FTM-TRS is widely recognized for its reliability in assessing tremor severity in clinical settings [32,48], yet its direct incorporation into machine learning models has not been thoroughly explored. This study demonstrates that coupling well-established clinical scales with advanced machine learning techniques can enhance the objectivity and precision of ET assessments.

While traditional applications of the FTM-TRS rely on subjective evaluation, incorporating it into a machine learning pipeline allows for automated, rater-independent classification. Existing research has largely focused on adapting clinical scales to sensor-based systems rather than embedding them into predictive models. For example, ref. [31] developed an accelerometry-based method to quantify tremor severity using FTM criteria, but their work was limited to translating clinical scales into sensor data without leveraging machine learning for automated classification. Similarly, ref. [47] reviewed intelligent devices for ET assessment but did not discuss integrating the FTM-TRS with machine learning models.

This pipeline holds promise for transforming ET monitoring and diagnosis in both clinical and telemedicine contexts. By leveraging objective and automated classification methods, the pipeline can assist clinicians in tracking disease progression and tailoring personalized treatment plans. However, implementing this approach in real-world settings requires further validation on larger and more diverse datasets that encompass the full spectrum of TRS severity. Expanding the dataset to include broader tremor severity representations and diverse patient populations would enhance the model’s generalizability and support its translation into clinical practice. Ultimately, integrating this pipeline into standard ET assessments could streamline clinical workflows, reduce reliance on subjective evaluations, and enable continuous, real-time monitoring of disease progression in telemedicine settings. This work highlights the need for collaborative efforts to validate and industrialize such pipelines, paving the way for more accessible and reliable tremor diagnosis and management.

## 5. Interpretability and Explainability

AI methodologies, especially machine learning/deep learning models, are increasingly used in health issues for diagnosis, prediction, treatment planning, and operational decision-making [49,50]. However, these models often become black boxes. Their internal performance is not evident, which creates barriers for clinicians, patients, and regulators in understanding how they make decisions. This creates significant challenges for producing useful and trustworthy technologies and systems for movement disorder diagnosis and management [51]. In this work, interpretability and explainability are therefore essential. The methodology we developed is designed to be explainable and easy to understand, intuitive, and accessible even to non-technical staff.

Firstly, using clinically interpretable TRS scores and geometric features, such as spiral radius and signal residues, enables intuitive analysis. These features link quantitative measures to observable phenomena, such as drawing stability or tremor regularity, enhancing the trustworthiness of models and results. Remarkably, they offer interpretability comparable to omics biomarkers, similar to the proteins or genes that are meaningful units in genomics. Spiral radius and signal residues represent concrete units of motor performance, making digital biomarkers transparent and accessible even to clinicians with standard statistical training.

Secondly, to enhance interpretability and align with clinical research practices, we integrated classical dimensionality-reduction techniques, such as PCA and LDA. Widely used in omics research, these methods reduce thousands of molecular variables into a few structured, interpretable components, allowing clinicians to visualize patient stratification, detect disease, and identify meaningful patterns. Applying PCA and LDA to time-series data replicates this familiar analytical paradigm, ensuring dimensionality reduction is intuitive, transparent, and clinically aligned.

Thirdly, managing false positives (FPs) and false negatives (FNs) is critical in medicine, as FPs can cause unnecessary treatments and FNs can delay diagnosis, requiring comprehensive evaluation metrics, especially in early disease stages or small datasets [52,53]. Common evaluation metrics, such as accuracy, measure overall correct predictions but can be misleading for imbalanced data. Precision, sensitivity (recall), and specificity assess the ability to correctly identify positives and negatives, maximizing true positives (TPs) and minimizing false negatives and false positives, respectively. Precision evaluates the reliability of positive predictions, and the F1-score balances precision and recall. ROC-AUC summarizes overall discriminative performance by combining specificity and sensitivity across thresholds.

Fourthly, from a traditional statistical perspective, methods such as linear regression provide clinicians with a statistical standard background and interpretable insights into how features influence predictions. When implemented in classical machine learning models, such as SVMs, interpretability can be preserved by linear-kernel SVMs, similar to multivariate regression, while kernel-based SVMs model complex relationships by mapping nonlinear separable classes while still allowing feature contribution assessment. In small datasets, carefully tuning SVM parameters, including the regularization constant and kernel settings, is essential to prevent overfitting, ensure generalization, and maintain explainable models.

In summary, the combination of geometric features, TRS scores, clinically oriented evaluation metrics, and classical statistical and machine learning models offers a robust, transparent, and clinically meaningful analytical framework. Finally, this approach balances minimizing misdiagnoses with generalizability to new patients, enabling the integration of machine learning tools into routine clinical practice. By leveraging SVM explainable models, widely used PCA and LDA methods, and interpretable features comparable to genes or proteins, high-dimensional motor data are reduced, interpreted, and translated in ways clinicians understand and trust.

## 6. Conclusions

### 6.1. Concluding Remarks

This study demonstrates the pipeline’s robust performance and clinical potential for ET diagnosis. By leveraging both spatial and frequency domain features, the pipeline consistently achieved high classification accuracy across multiple validation techniques, underscoring its robustness and adaptability.

The integration of LDA significantly enhanced class separability, ensuring reliable classification even in high-dimensional spaces. These findings validate the pipeline’s suitability for real-world tremor classification tasks, including clinical and telemedicine-based deployment. The ability of DCT residues to reflect localized disruptions in spiral data complements the global movement patterns captured by radius features, aligning with clinical understandings of tremor-induced irregularities. This orthogonality provides a strong foundation for combining spatial and frequency domain features, enhancing the diagnostic power and robustness of automated classification pipelines. The consistent performance across both feature types highlights the utility of handwriting-based diagnostics for ET and related disorders, offering a non-invasive, low-cost alternative to resource-intensive methods like neuroimaging.

### 6.2. Future Directions

To address these challenges, future work should explore the application of deep learning models, such as Convolutional Neural Networks (CNNs) or Recurrent Neural Networks (RNNs), to process raw time-series data directly. These models can automatically extract relevant features, removing the need for manual feature engineering, and provide end-to-end solutions with reduced complexity. This approach could be used for the following:Simplifying Workflows: Eliminating preprocessing steps like DCT, PCA, or LDA, streamlining the pipeline.Improving Scalability: Reducing computational costs, making the solution more accessible for industrial and telemedicine applications.Enhancing Generalization: Leveraging large datasets to train robust models capable of capturing nuanced patterns beyond the engineered features.Facilitating Deployment in Telemedicine: Deep learning models integrated into edge devices or cloud-based platforms can enable real-time classification, making tremor analysis feasible in remote clinical settings.

Further research should also explore multimodal analysis, integrating features such as neuroimaging (e.g., fMRI) with handwriting biomarkers to enhance diagnostic precision. Advanced frequency domain transformations (e.g., Wavelet Transform) and fusion techniques could provide deeper insights into tremor severity and variability. Finally, validation on larger and more diverse datasets is essential to establish the pipeline’s generalizability and ensure its real-world applicability. By addressing these challenges and expanding the scope of analysis, this work paves the way for scalable, non-invasive diagnostic tools that can transform ET management into clinical practice and telemedicine. Future developments in deep learning and multimodal integration will further enhance the accessibility and efficacy of tremor diagnosis, advancing the field toward precision healthcare solutions.

## Figures and Tables

**Figure 1 sensors-26-00244-f001:**
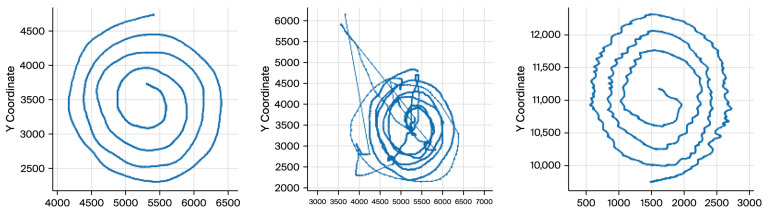
The progressive loss of drawing consistency as tremor severity increases.

**Figure 2 sensors-26-00244-f002:**
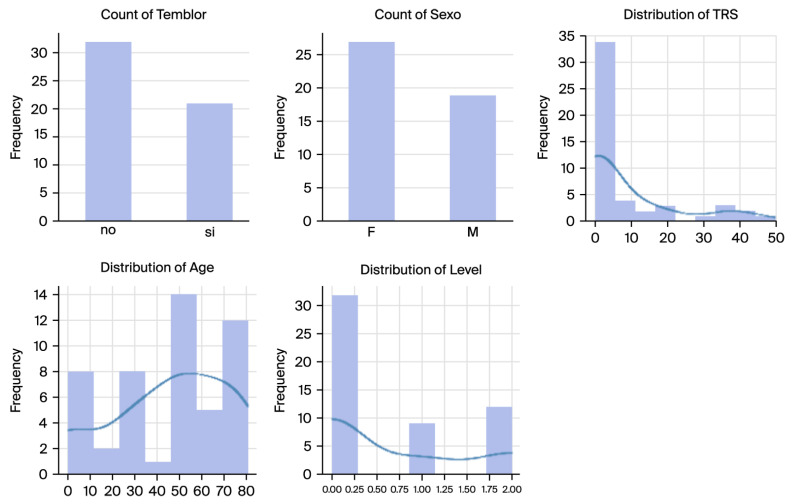
Distributions showcasing the dataset’s variability and balance of demographic and clinical factors.

**Figure 3 sensors-26-00244-f003:**
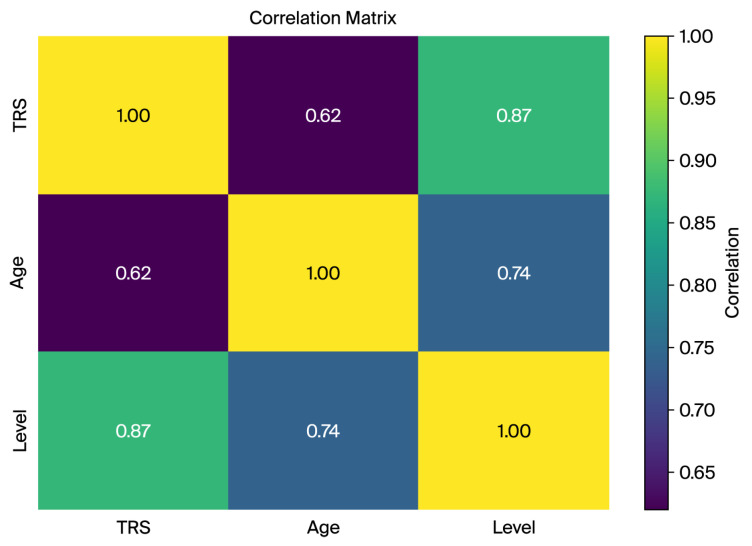
The correlation matrix of key dataset features (TRS, age, and tremor level) indicating strong associations between tremor severity and derived tremor level groups.

**Figure 4 sensors-26-00244-f004:**
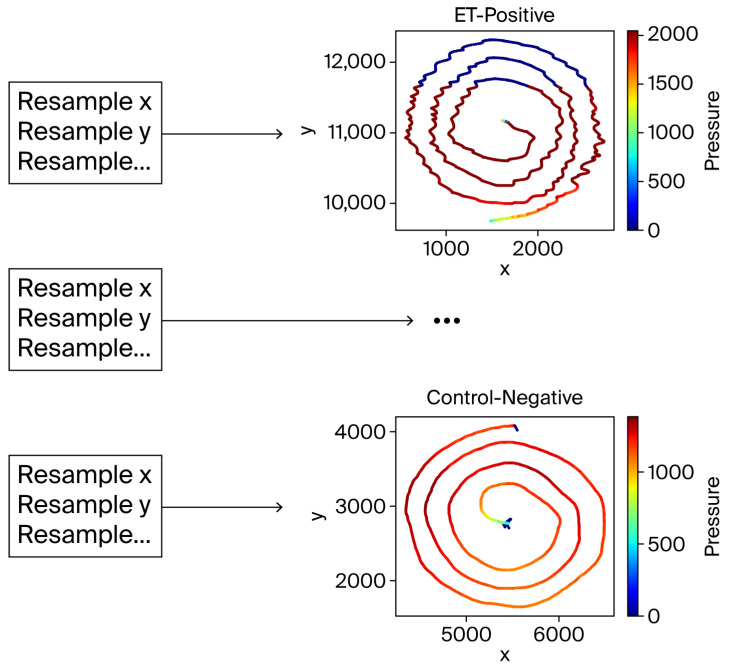
The resampling process. Despite interpolation, the spiral shapes remained visually intact, validating the resampling approach.

**Figure 5 sensors-26-00244-f005:**
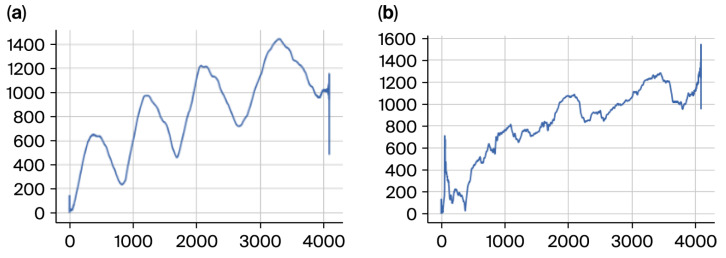
(**a**) Negative subject radius to the left and (**b**) positive to the right.

**Figure 6 sensors-26-00244-f006:**
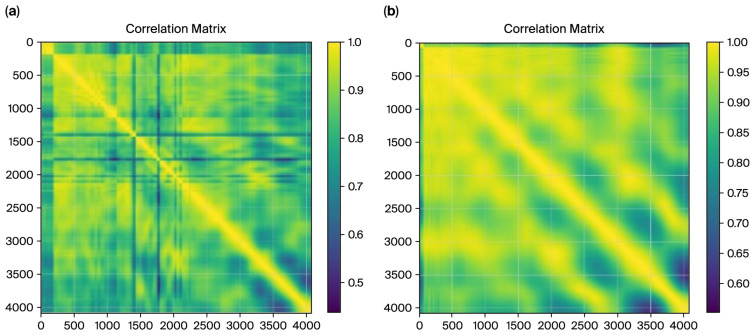
The correlation matrix, from radius, for ET cases (**a**), and the matrix for control cases (**b**).

**Figure 7 sensors-26-00244-f007:**
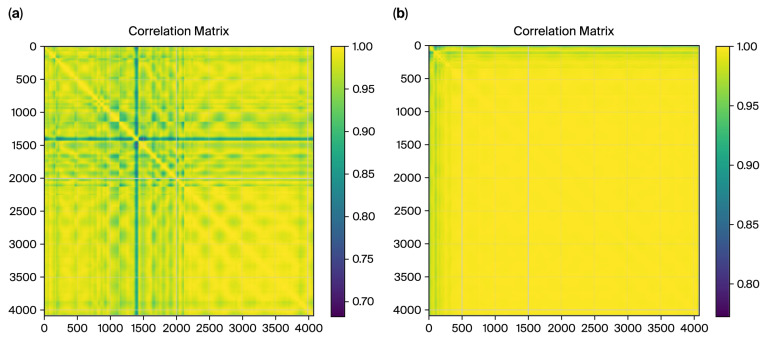
The correlation matrix from DCT residues for ET cases (**a**), and the matrix for control cases (**b**).

**Figure 8 sensors-26-00244-f008:**
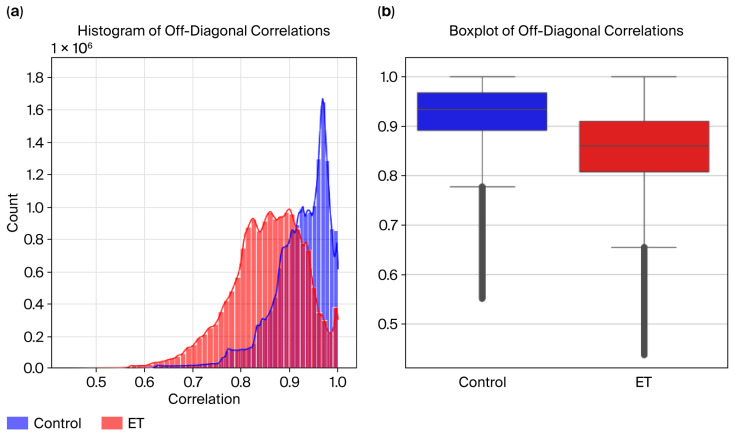
(**a**) Radius histogram and (**b**) boxplot.

**Figure 9 sensors-26-00244-f009:**
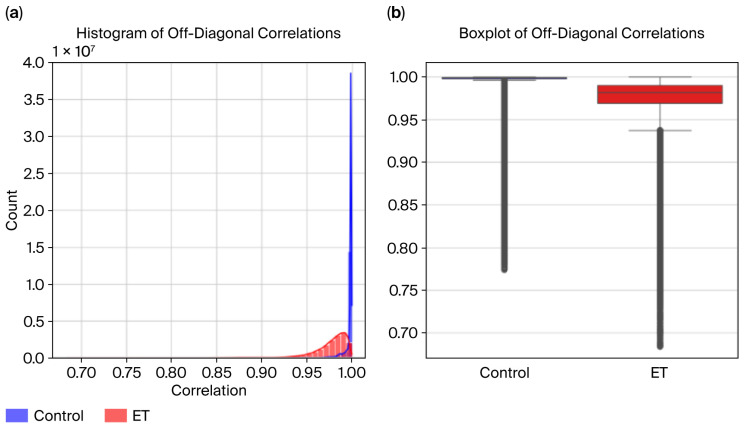
DCT residues: (**a**) histogram and (**b**) boxplot.

**Figure 10 sensors-26-00244-f010:**
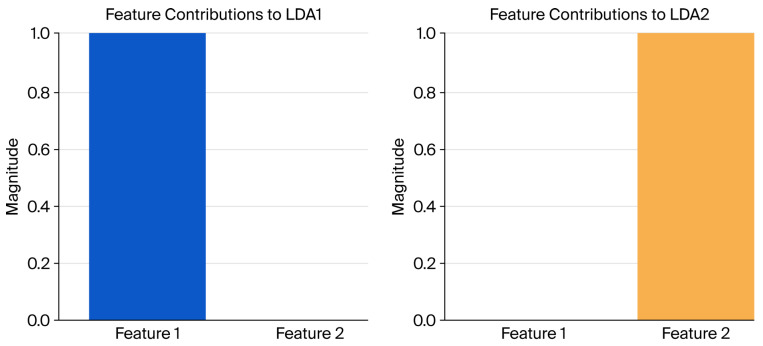
LDA feature contributions.

**Figure 11 sensors-26-00244-f011:**
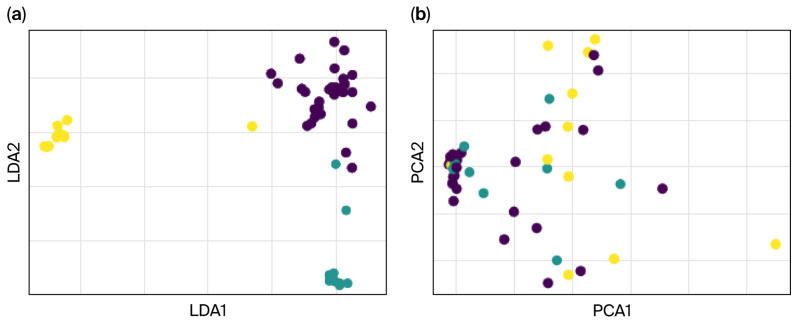
The effective separation between control (dark) and tremor severity groups (light), highlighting the utility of (**a**) LDA and (**b**) PCA for maximizing class separability.

**Figure 12 sensors-26-00244-f012:**
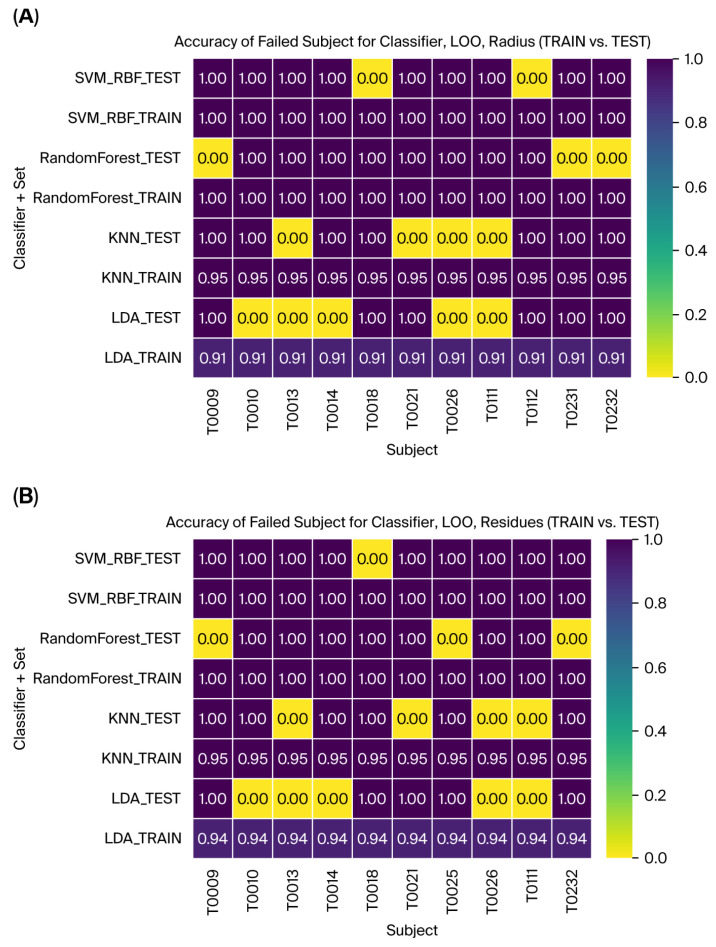
The failed subjects with Leave-One-Out validation for all the classifiers and TRAINI and TEST set for (**A**) Radius and (**B**) Residues.

**Table 1 sensors-26-00244-t001:** Classification methods, parameters, and range of configuration for generalization.

Method	Key Parameters	Description
SVM	- C = 0.1–1. - gamma = 0.1. - Hyperparam.: Grid Search.	Handles nonlinear separability. Optimized to balance training error and model generalization.
k-NN	- k = 5–10. - Euclidean Distance.	Provides optimal balance between sensitivity and specificity.
RF	- *n* = 100 DT. - Hyperparam.: Grid Search.	Captures nonlinear patterns. Reduces overfitting and improves generalization through random splits.
LDA	- Linear combinations of features.	Effective for linearly separable data. Used both as a classifier and for feature reduction.

**Table 2 sensors-26-00244-t002:** Model performance comparison using Leave-One-Out validation for radius.

v	Train Score (%)				Test Score (%)			
Model	Acc	Pre	Rec	F1	Acc	Pre	Rec	F1
SVM (RBF Kernel)	100.00	100.00	100.00	100.00	96.23	95.56	93.33	93.92
Random Forest	100.00	100.00	100.00	100.00	94.34	92.66	90.77	91.63
k-NN (k = 5)	97.68	98.69	95.90	97.14	92.45	96.08	86.67	89.58
LDA (Classifier)	90.89	89.91	85.42	87.34	90.57	89.56	84.87	86.86

**Table 3 sensors-26-00244-t003:** Model performance comparison using Leave-One-Out validation for DCT residues.

	Train Score (%)				Test Score (%)			
Model	Acc	Pre	Rec	F1	Acc	Pre	Rec	F1
SVM (RBF Kernel)	100.00	100.00	100.00	100.00	98.11	97.62	96.67	97.01
Random Forest	100.00	100.00	100.00	100.00	94.34	95.54	90.00	91.81
k-NN	95.94	97.77	92.82	94.82	92.45	96.08	86.67	89.58
LDA (Classifier)	94.12	92.72	91.12	91.66	90.57	89.56	84.87	86.86

**Table 4 sensors-26-00244-t004:** Performance summary of the PCA → LDA → SVM pipeline across various validation approaches, including hold-out validation, cross-validation, stratified K-Fold, and robustness testing under Gaussian noise perturbations.

Method	Train Score (%)				Test Score (%)			
PCA → LDA → SVM	Acc	Pre	Rec	F1	Acc	Pre	Rec	F1
Radius								
SVM-Hold-out	100.00	100.00	100.00	100.00	81.82	53.33	66.67	58.33
SVM-5-CV	100.00	100.00	100.00	100.00	98.18	96.67	98.33	96.83
SVM-St-CV	100.00	100.00	100.00	100.00	98.00	97.78	96.67	96.44
SVM-Noise R	100.00	100.00	100.00	100.00	100.00	100.00	100.00	100.00
DCT residues								
SVM-Hold-out	100.00	100.00	100.00	100.00	90.91	91.67	83.33	84.13
SVM-5-CV	100.00	100.00	100.00	100.00	98.18	96.67	98.33	96.83
SVM-St-CV	100.00	100.00	100.00	100.00	100.00	100.00	100.00	100.00
SVM-Noise R	100.00	100.00	100.00	100.00	100.00	100.00	100.00	100.00

## Data Availability

The datasets generated by and/or analyzed during the current study are not publicly available due to ethics and privacy requirements, but they are available from the corresponding authors upon reasonable request. All the code is open source upon reasonable request. The code is available at GitHub (https://github.com/spolex/spiral, accessed on 25 December 2025).

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
