# Peer review of "Essential Tremor Severity Assessment Using Handwriting Analysis and Machine Learning"

_sensors, 2025, doi:10.3390/s26010244_

Round 1

Reviewer 1 Report

Comments and Suggestions for Authors

The paper in review is devoted to an interesting and actual problem which may attract the interest on behalf of the audience of the journal. The paper is clearly written and easy to follow.  On the level of data acquisition and following analysis the reviewer did not find and sever issues besides Table 4 where all the values are identical and suspiciously high which lead high margin separability between the classes? This looks very suspicious. Also in previous tables some model quality metrics reported to be 100 which is rather strange. 

The main points of the critique are as follows:
1. Introduction and positioning of the paper. In the opinion of the reviewer, the authors did not identify any research gaps or inconsistencies among previously published results. Introduction just lists some results without discussing its goodness, drawbacks, etc, and no numeric values are presented.  Therefore, on the level of justification and motivation, it is unclear why the current research is necessary, what is the novelty of the results, what the benefits of the proposed approach and how it relates to previously published concepts.  Moreover, spiral drawing test is one of the most studied once, and a lot of methods were developed for its analysis. It is surprising that the authors do not use or compare their approach with that of others. Pullman (1998) proposed using digital tables in their paper "Spiral analysis: a new technique  for measuring tremor with a digitizing tablet." Later Seminal paper of Drotar appeared "Evaluation of handwriting kinematics and pressure for differential diagnosis of Parkinson's disease", Valla has studied the application for supervised selection in "Evaluation of handwriting kinematics and pressure for differential diagnosis of Parkinson's disease", and one can continue this list. Please, note that this is not a citation request (the authors can identify which results they do need). Since the authors use Statistical machine learning classifiers, at least some sort of attempt to use previously developed techniques and results, or at least discussion is expected. Also, it would be nice to have a separate section fully devoted to the literature review. 

2. Formatting of the paper. Immediately on the first page "What are the main findings? 18 ï‚· First bullet." Please proof read the paper properly.  Presented figures and diagrams are in raster graphic which negatively affect their quality, please generate proper vector format graphs.

Comments on the Quality of English Language

Proof reading is required.

Reviewer 2 Report

Comments and Suggestions for Authors

1. The template guidelines in the Abstract should be removed.
2. Justify that a sampling frequency of 100 Hz is appropriate.
3. The number of references should be increased. Twenty is too few.
4. There should be a table with the results of each patient's evaluation.
5. The analysis of the results of the machine learning model metrics should be increased.
6. A section on the interpretability and explainability of the proposed method should be added to the paper.

Reviewer 3 Report

Comments and Suggestions for Authors

This paper focuses on the non-invasive diagnosis and severity classification of Essential Tremor (ET). The writing is generally well-structured. The authors propose a machine learning diagnostic pipeline based on Archimedes spiral handwriting data, combining spatial-domain features (radius) with frequency-domain features (DCT residues), and introducing the clinically relevant Fahn-Tolosa-Marin Tremor Rating Scale (TRS) for severity labeling and feature information. The authors further apply PCA + LDA for dimensionality reduction and interpretability enhancement, and validate the effectiveness of the method through a comparison of multiple models (SVM, k-NN, LDA, Random Forest) and various validation strategies (LOOCV, K-fold, stratified K-fold, robustness with noise perturbations). The paper not only provides a novel approach for integrating clinical scales into automated handwriting analysis, but also presents a feasible, low-cost telemedicine application, which has practical value for improving ET screening and follow-up evaluation. Overall, the article demonstrates innovation in research perspective, feature construction and interpretability design, as well as potential clinical/telemedicine deployment.

   On one hand, this paper is generally well-written, and I am confident that the authors have conducted a thorough and in-depth analysis of tremor patterns in handwriting trajectories as well as the enhancement of class separability and interpretability through PCA → LDA. On the other hand, I have also identified some inconsistencies in the presentation of the content and certain shortcomings in the experimental setup. Revisions are needed before publication to meet the journal's standards. I will provide a more detailed explanation below.

   1.The table data format should be standardized throughout the manuscript. All tables should be unified into a three-line table format to enhance aesthetics and readability. Additionally, the data related to the proposed method can be bolded in the tables to increase contrast and highlight key information.

   2.Update the references related to non‑comparative methods. Include more relevant works published after 2020, so as to ensure that the literature review is both comprehensive and up-to-date.

   3.The images are not clear enough. The text in Figures 1 to 11 is small, and I recommend enlarging the text in the images and uploading high-resolution versions to improve readability and clarity of the figures.

   4.There is ambiguity in the dataset description that needs further clarification. In line 88–94, it is mentioned that "50 ET + 50 control participants" were recruited. However, in line 110–113, it states "The BIODARW version includes 53 samples (24 ET / 29 Control)." Please clarify the actual number of samples used for modeling and resolve the discrepancy in the description within the text.

   5.Check if the parameter settings are reasonable. In line 303, the SVM's C value is set to 10^10, which may lead to a hard margin and cause overfitting during training. If this parameter was derived from data issues or optimization algorithms (such as cross-validation or grid search), please provide an explanation of the rationale for the parameter choice in the text.

   6.The p-value is incorrect. Theoretically, the p-value should not be 0; it is likely that your software rounded it off during the calculation. To maintain scientific rigor, you should change the p-value to p < 0.001 to indicate statistical significance.

   7.Optimize the abstract content. It is recommended to include specific test metrics in the abstract, such as accuracy, F1 score, etc., to more clearly present the experimental results and the performance of the model.

   8.The 100% accuracy result is overly ideal. In lines 476, 489, 511, 513, 514, 518, 519, it is repeatedly mentioned that the accuracy reaches 100% under different settings (such as 5-fold, KNN, and various noise perturbation conditions). This is uncommon in small sample sizes (only 53 data points). It is recommended that the authors carefully check for potential data leakage, such as TRS being used both as a numerical/categorical feature in the model and as a label for classification, which could lead to label leakage, or the possibility of the same subject’s samples appearing in both the training and test sets, leading to subject leakage. If these issues are ruled out, no changes are necessary.

   9.The paper contains some language and grammar errors. It needs to be refined to ensure it meets the journal's terminology requirements and to improve the overall quality of the manuscript.

Round 2

Reviewer 2 Report

Comments and Suggestions for Authors

The authors addressed all the comments, and the work was improved.